# Research on Real-Time Detection of Maize Seedling Navigation Line Based on Improved YOLOv5s Lightweighting Technology

**Hailiang Gong, Xi Wang * and Weidong Zhuang**

College of Engineering, Heilongjiang Bayi Agricultural University, Daqing 163319, China; ndhailiang@byau.edu.cn (H.G.); zhuangwd@byau.edu.cn (W.Z.)

**\*** Correspondence: ndwangxi@byau.edu.cn

**Abstract:** This study focuses on real-time detection of maize crop rows using deep learning technology to meet the needs of autonomous navigation for weed removal during the maize seedling stage. Crop row recognition is affected by natural factors such as soil exposure, soil straw residue, mutual shading of plant leaves, and light conditions. To address this issue, the YOLOv5s network model is improved by replacing the backbone network with the improved MobileNetv3, establishing a combination network model YOLOv5-M3 and using the convolutional block attention module (CBAM) to enhance detection accuracy. Distance-*IoU* Non-Maximum Suppression (*DIoU*-NMS) is used to improve the identification degree of the occluded targets, and knowledge distillation is used to increase the recall rate and accuracy of the model. The improved YOLOv5s target detection model is applied to the recognition and positioning of maize seedlings, and the optimal target position for weeding is obtained by max-min optimization. Experimental results show that the YOLOv5-M3 network model achieves 92.2% mean average precision (*mAP*) for crop targets and the recognition speed is 39 frames per second (*FPS*). This method has the advantages of high detection accuracy, fast speed, and is light weight and has strong adaptability and anti-interference ability. It determines the relative position of maize seedlings and the weeding machine in real time, avoiding squeezing or damaging the seedlings.

**Keywords:** maize seedlings; autonomous navigation; deep learning; crop row detection; inter-row weeding

## 1. Introduction

The early growth stage of maize is a crucial time for weed management. While excessive use of chemical herbicides is effective, it may damage the integrity of soil and water [1]. With the advancement of agricultural technology, there is an urgent need for innovative weed management solutions that not only effectively suppress weeds [2], but also reduce labor, increase efficiency, and enhance crop yields [3]. Mechanical weeding serves as a viable alternative to chemical pesticides, saving labor and time compared to manual methods [4,5]. However, traditional mechanical weeding often struggles to accurately identify and locate maize seedlings, leading to potential damage [6,7]. This issue primarily stems from the diverse and uneven distribution of weed species in the field as well as the need to improve the recognition accuracy and robustness of existing methods in complex backgrounds [8]. These methods currently cannot diagnose the relative position of crops and weeding machinery in real time, making it difficult to precisely control weeding locations [9]. Therefore, developing a method that can accurately identify and precisely locate maize seedlings to provide reference positions for inter-row weeding can improve the weeding efficiency.

Machine learning has been integrated into mechanical weeding, improving detection accuracy and reducing manual labor costs [10]. For large-scale inter-row weed management, the accuracy of inter-row detection, weed identification, and automatic

control is crucial [11]. Despite advancements such as RTK-GPS navigation, the effectiveness of existing methods is challenged by variations in plant spacing caused by environmental factors and the accuracy of seeders [12,13]. Machine vision combined with precision control systems has been proposed to address these challenges [14], utilizing image processing and sensor technology to reduce labor intensity and costs [15]. In recent research on crop row detection, a central line representing the width of the crop row in the image is usually extracted as the navigation information for mechanical travel [13]. Most methods can be divided into two steps. The first step is to accurately segment the image and background to extract the green plants in the image, including weeds and crops to obtain a binary clipping image. The second step is to extract the crop rows from the binary image. The process of crop row extraction is generally based on the model [16]. Ref. [17] employed a horizontal strip method to obtain crop row features and determined the final clusters using a location clustering algorithm and the shortest path method. However, if there are weeds near the crop rows, the accuracy of the horizontal strip method is poor. Ref. [18] proposed a new method for detecting wheat rows using a mobile window to scan images of early-stage wheat, and crop row feature points were obtained using the Hough transform and vanishing points. Meanwhile, Ref. [18] introduced a new method for detecting curved and straight crop rows using a Hough transform-based approach that overcomes the effects of varying lighting conditions and different plant heights and volumes during different growth stages. Furthermore, Ref. [19] proposed an improved multi-ROI method to accurately detect crop rows in complex field environments.

With the emergence of computer vision and artificial intelligence, convolutional neural networks (CNNs) have gained widespread attention in the field of computer vision [20]. Advanced agricultural countries have already made significant strides in the development of agricultural robots [21]. CNNs have shown their versatility and prowess in various intelligent machine learning applications, and recently they have been applied to weed control [22]. For instance, Ref. [23] proposed a method that combines optical flow with CNNs to achieve accurate crop row segmentation in weed-infested fields. Moreover, Ref. [24] introduced a weed species identification method using the CNN-based approach in wheat fields, with an accuracy rate of 97%. Similarly, Ref. [25] developed a soybean weed detection model using CNNs, which achieved an accuracy of 98% based on images from multiple soybean fields. Additionally, Ref. [26] proposed an optimized method combining dilated convolution with global pooling for crop and weed recognition with excellent recognition results. Ref. [27] further improved the Xception model by employing exponential linear units as activation functions and a lightweight convolutional neural network-based weed recognition model, which attained an average test recognition accuracy of 98.63% across eight weed species and young maize. Finally, Ref. [28] developed a field navigation agricultural robot that can track early crop rows in unstructured and irregular agricultural environments by proposing a row anchor selection classification method.

As the demand for deep learning algorithms on mobile devices has increased, research on lightweight network structures such as MobileNet and other designs have also become popular [29]. These networks improve running speed while maintaining accuracy, reducing requirements for parameters and computing power, and are suitable for embedded and mobile devices in real-life scenarios. Furthermore, to meet the demands of deep learning, the performance and computing power of embedded devices have also improved, as indicated in recent studies. In terms of object detection, single-stage models such as SSD, EfficientDet, RetinaNet, and YOLO series can directly localize and classify targets [30]. Recently, Ref. [31] proposed an optimized method for cotton seedling weed identification and localization based on Faster R-CNN, achieving an average recognition accuracy of 88.67% for cotton seedlings and weeds. Meanwhile, Ref. [32] applied depthwise separable convolution and a residual structure while incorporating an attention mechanism into the YOLOv4 feature extraction backbone network to detect

weeds in carrot fields, providing a promising solution for accurate weed detection in agricultural applications. Ref. [16] utilized the YOLOv5 network model to extract regions of interest (ROIs) from images, using these feature points to detect driving areas, thereby addressing the impact of background clutter on detection results. The average errors in calculating the driving path and heading angle were 2.74°, meeting the real-time and precision requirements of agricultural machinery visual navigation. Ref. [4] proposed a new spatial pyramid pooling structure, ASPPF, and constructed the ASPPF-YOLOv8s network model, achieving a detection accuracy of over 90% for the maize plant heart. However, the detection is performed individually for each weed, which may result in false detections when dealing with a large variety and dense distribution of weeds in the field. To address these challenges, Ref. [33] compared the performance of several lightweight deep learning models to minimize the model size and compute load while ensuring accuracy. The results show that MobileNetV2 and ShuffleNetV2 performed better than other models in both efficiency and effectiveness.

Research both domestically and internationally has predominantly focused on the following aspects: (1) methods based on morphological features, however the identification performance is unsatisfactory in complex environments; (2) the use of deep learning object detection models, with the need to improve the accuracy and robustness of small object recognition; (3) utilizing improved backbone networks for feature extraction, yet the effectiveness in maize seedling scenes remains to be verified; (4) employing single-stage object detection models, which offer high recognition efficiency, but limited capability in scenarios involving overlapping targets. Overall, the current methods still require improvement in the identification and positioning of maize seedlings in complex backgrounds. To address this issue, this paper proposes a maize seedling automatic recognition and navigation positioning method based on an improved YOLOv5s object detection model. Specifically, this study adopts lightweight backbone networks such as MobileNetV3 for feature extraction and incorporates the Convolutional Block Attention Module (CBAM) to enhance small object recognition capability. Simultaneously, the method utilizes the least squares fitting to extract precise navigation centerlines and obtains the target position of the weeding wheel through max-min optimization, providing reference for mechanical weeding. Experimental results demonstrate that this method significantly improves recognition accuracy while maintaining speed, effectively resolving the issue of identification and positioning in complex environments.

## 2. Materials and Methods

### 2.1. Structure and Operating Principle of Inter-Row Weeding Machine

The intelligent inter-row weeding mechanism consists of an image recognition system, a control system, a hydraulic system, and a mechanical structure. The mechanical structure mainly includes a beam frame structure, an inter-row weeding mechanism, a lateral displacement mechanism, and a steering mechanism. It can automatically align with the navigation line, improve weeding efficiency, and reduce plant injury. The overall structure is shown in Figure 1.

The intelligent inter-row weeding equipment extracts real-time seedling and weed information from the field through a camera, performs image recognition processing to accurately distinguish seedlings, weeds, and soil, and locates the positions of seedlings. Using maize seedling crop rows as a reference benchmark, computer vision technology is introduced to fit the center lines of two crop rows, and calculate the middle position between the two lines as a guide line. The lateral movement of the weeding machine is controlled and adjusted using the guide line as a reference. The weeding machine's execution mechanism automatically adjusts its lateral position as the vehicle deviates, maintaining a safe distance from the crop rows. The computer processes relevant information and sensors feedback, issues control signals, and drives the weed removal equipment to complete tasks such as lateral movement, turning, weed removal, and seed avoidance. The hydraulic system outputs control signals, and the hydraulic cylinder, as the actuator of the closed-loop

control system, adjusts the lateral displacement of the weeding wheel accordingly to avoid damaging the maize seedlings. The schematic diagram in Figure 2 illustrates the weed control principle of the inter-row weeding machinery.

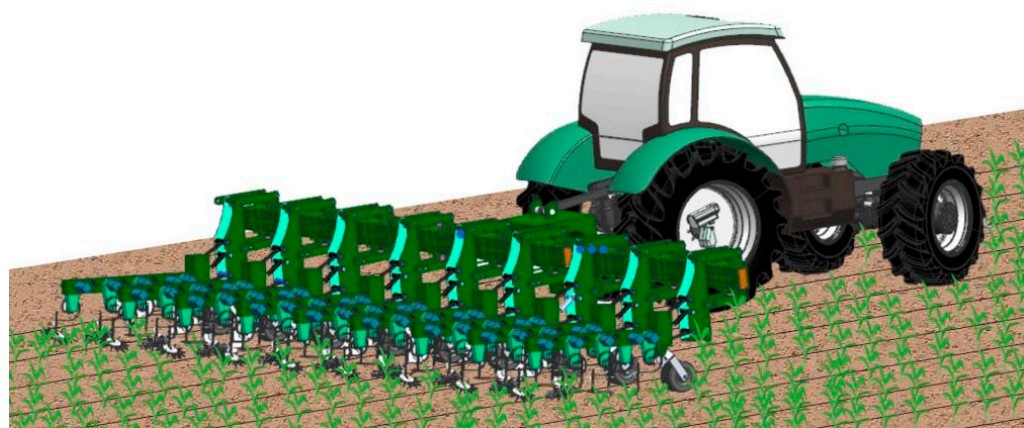

**Figure 1.** The overall structure of the crop inter-row weeding machinery.

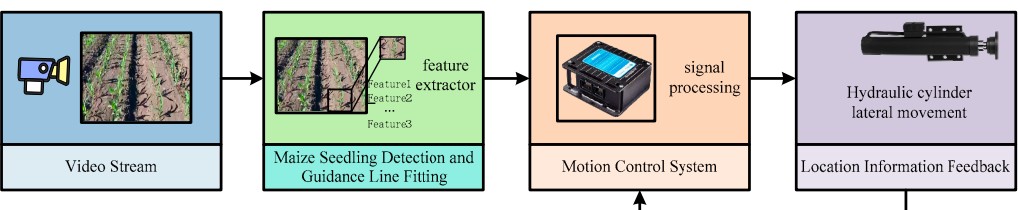

**Figure 2.** The schematic diagram of inter-row weeding machinery's weed control principle.

*2.2. Method for Extracting Crop Row Navigation Line*

This study proposes a method for extracting the navigation line of crops during the seedling stage of maize cultivation. The method is divided into three stages:

Using deep learning, a maize detection model for crop row maize seedling target detection is trained. Firstly, the acquired RGB image of the crop row is analyzed according to the camera pitch angle and the imaging relationship of the seedlings, and the ROI region in the image is constructed. Then, based on the improved YOLOv5s target detection model, the maize crop targets are detected, and the maize category, detection boundary box, and category confidence are output. As shown in Figure 3a, the red boundary box surrounds the detected maize seedlings, showing their classification and stability.

As shown in Figure 3b, the dots in the bounding box represent the position of the detected maize seedlings. This step is to calculate the positions of the seedlings and represent them with coordinates. The geometric center of the bounding box can represent the position of the maize seedlings, as they are uniformly distributed around the plant stem during image acquisition. For the maize seedling line fitting, in the two crop rows in the ROI processing area, the coordinates of the maize seedlings in the crop rows were extracted, and linear fitting was adopted for the seedling target detection coordinates, as shown in Figure 3c. In this step, crop row bending diagnosis and guide line extraction were achieved. As the ultimate goal is to automatically control the lateral offset of the weeding mechanism to prevent them from damaging the seedlings, the optimal position of the weeding mechanism was calculated. In Figure 3d, the red midline is marked as the trajectory of the optimal position as the navigation line.

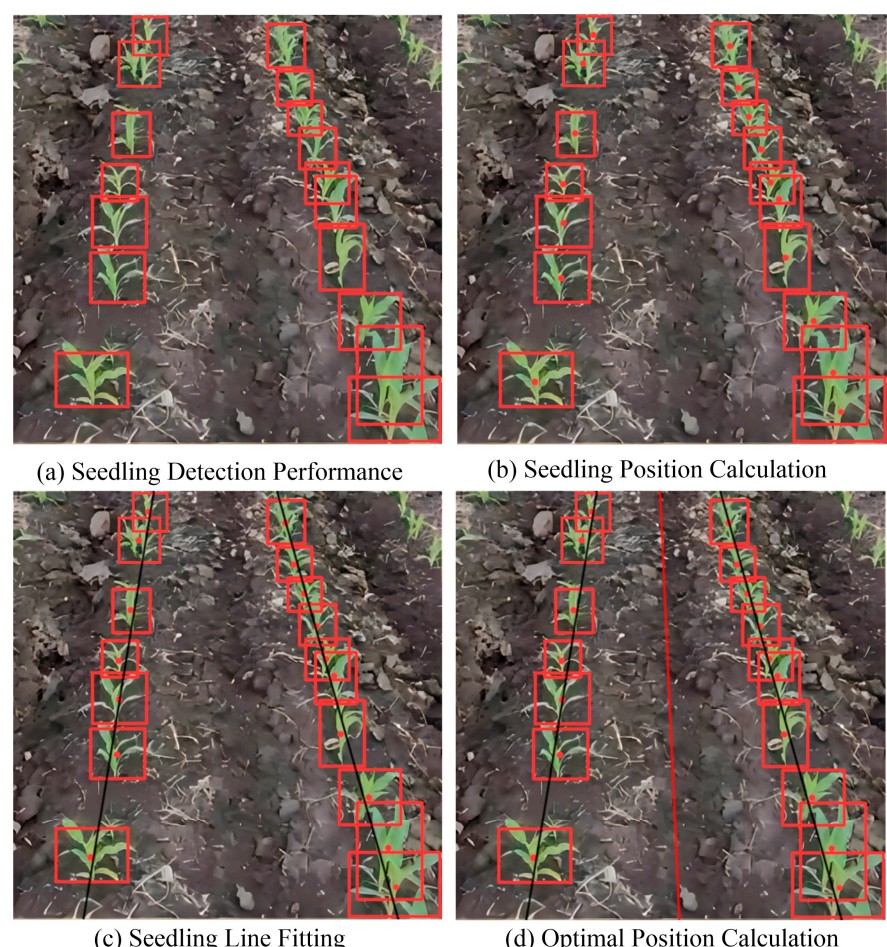

(a) Seedling Detection Performance      (b) Seedling Position Calculation

(c) Seedling Line Fitting      (d) Optimal Position Calculation

**Figure 3.** Results of the algorithm step.

### 2.3. YOLOv5 Object Detection Model

YOLOv5 is a one-stage detection model released by UitralyticsLLC, whose core idea is to take the whole picture as the network input, integrate the target decision and target recognition into one, and directly regress the position of the prediction box and the class of the prediction box at the output layer, with the advantages of smaller mean weight file, shorter training time, and inference speed [34,35]. The detection performance has been further improved. The YOLOv5 target detection network model structure has YOLOv5n, YOLOv5s, YOLOv5m, YOLOv5l, and YOLOv5x; these five models have similar structures, among them YOLOv5s network parameters are the smallest, training speed is the fastest, but AP accuracy is the lowest. If the detection target is mainly large-scale target and the training target pursues speed, YOLOv5s can meet the training conditions. The other four networks continue to deepen and widen the network on this basis, and the AP accuracy continues to improve, but the speed consumption is also increasing. Therefore, this paper decided to improve YOLOv5s, so that the improved model can not only maintain its speed advantage when applied, but also improve its accuracy, so as to be able to quickly and effectively train small targets.

The structure of the YOLOv5s model consists of four parts: a convolutional network-based backbone main network [30], which mainly extracts feature information of the image; a head detection head, which mainly predicts the target box and predicts the target class; a neck between the main network and the detection head; and a prediction layer that outputs the detection result, predicting the target detection box and label category, as shown in Figure 4.

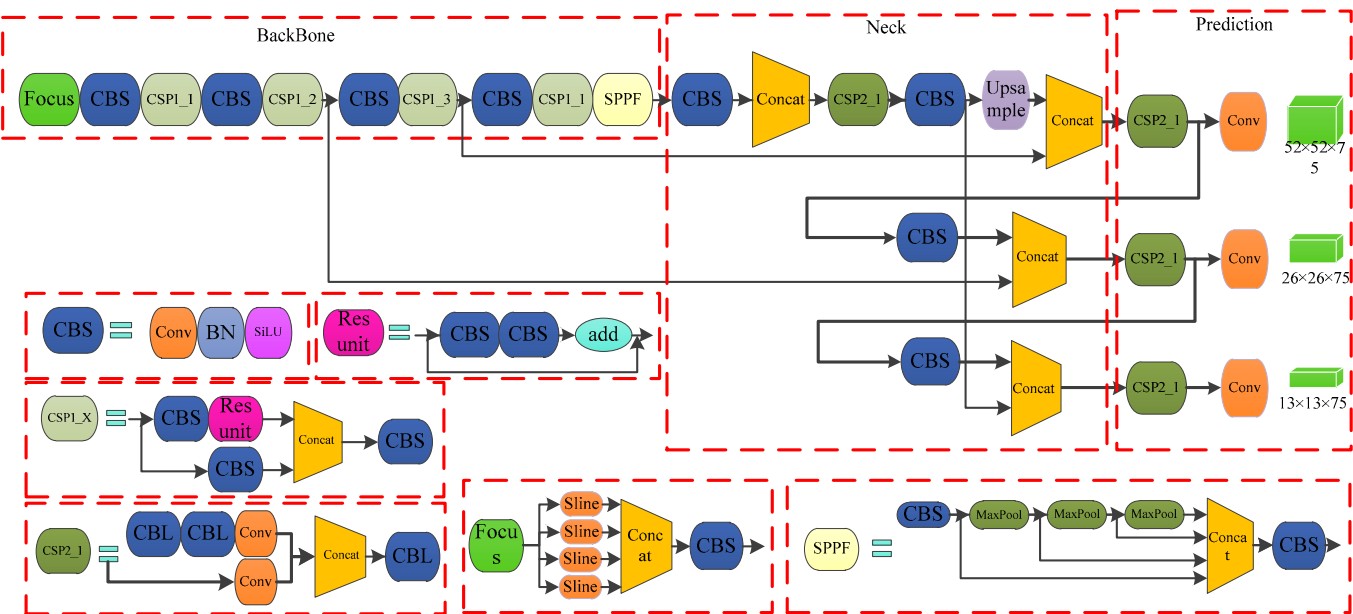

**Figure 4.** YOLOv5s structure diagram.

## 2.4. Improvements to YOLOv5's Object Detection Model

### 2.4.1. Design of YOLOv5-M3 Network Model

In the original YOLOv5s model, the CSPDarknet-53 network structure contains 52 standard convolution layers and 1 fully connected layer, which has many layers, high model complexity, and difficult training. Therefore, based on the original YOLOv5s, a lightweight real-time object detection neural network model YOLOv5-M3 was proposed to improve the inference speed of the model, and the CBAM attention mechanism network structure was integrated, as shown in Table 1.

**Table 1.** MobileNetv3 network model architecture parameters with attention mechanism.

| Floor | Input | Output | Numbers | Activation Function | CBAM Attention |
|---|---|---|---|---|---|
| Conv2D_BN_ hard-swish | $416^2 \times 3$ | $208^2 \times 16$ | 1 | hard-swish | $\times$ |
| Bneck_block | $208^2 \times 16$ | $208^2 \times 16$ | 1 | relu | $\times$ |
| Bneck_block | $208^2 \times 16$ | $104^2 \times 24$ | 2 | relu | $\times$ |
| Bneck_block | $104^2 \times 24$ | $52^2 \times 40$ | 3 | relu | $\checkmark$ |
| Bneck_block | $52^2 \times 40$ | $26^2 \times 112$ | 6 | hard-swish | $\checkmark$ |
| Bneck_block | $26^2 \times 112$ | $13^2 \times 160$ | 3 | hard-swish | $\checkmark$ |

YOLOv5-M3 is an end-to-end detection framework based on regression idea, and the network model is shown in Figure 5. MobileNetv3 is used as the backbone network of YOLOv5s to extract features, and the performance and running time of the model are studied. MobileNetv3 is improved to design the model lightweight, and CBAM attention mechanism is used to replace SENet module in the network model to optimize the accuracy of target detection and strengthen the focus on the detection target, thus reducing the decline of detection accuracy caused by the complex environment. The designed YOLOv5-M3 network model uses depthwise separable convolution to replace standard convolution, further reducing the model complexity, improving training efficiency and inference speed.

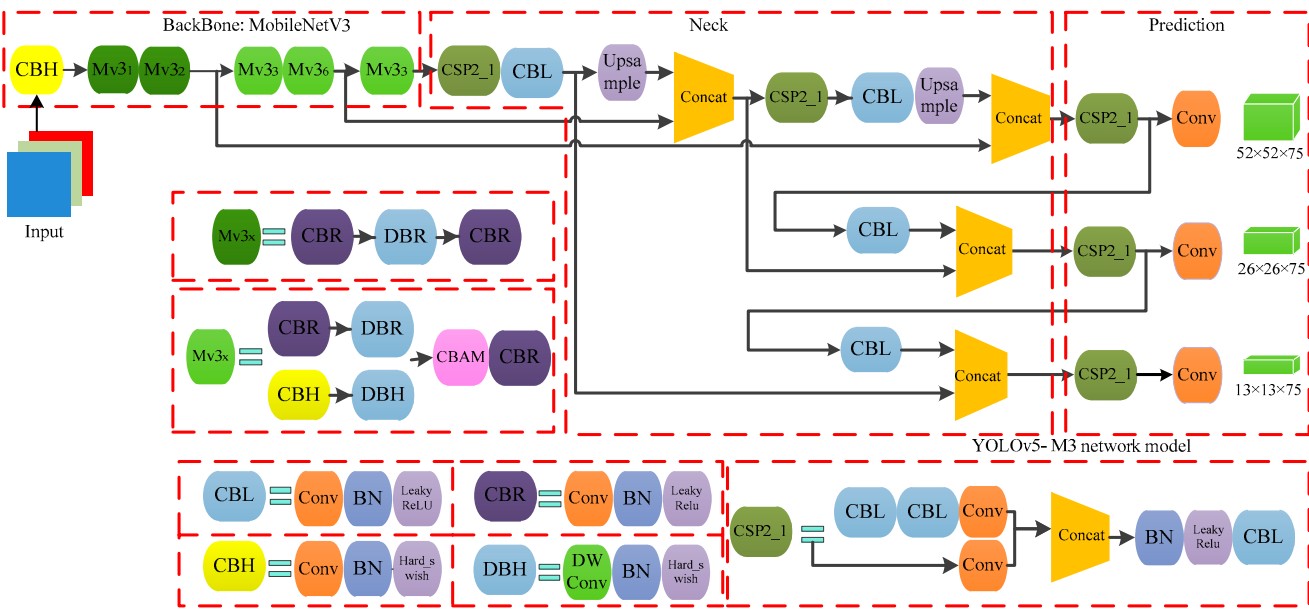

**Figure 5.** YOLOv5-M3 network model structure.

### 2.4.2. Optimization of Backbone Network in YOLOv5s

The CSPDarkNet53 backbone network model used in YOLOv5s introduces Cross Stage Partial Network (CSPNet) to extract effective deep feature information [36,37]. However, it was found in the experiment that only by directly adjusting the width multiplier and depth multiplier to lighten the model (when the width coefficient and depth coefficient are smaller than 1.0), there was a serious problem of missed targets in the video image. Therefore, when considering adjusting the lightened model, a lighter backbone network with a stronger feature extraction ability on the mobile side needs to be introduced. Replacing CSPDarkNet53 with the lightweight backbone network MobileNetV3 attempts to achieve a balance between lightness, accuracy, and efficiency.

MobileNetV1 is a lightweight CNN suitable for deployment on edge devices, which can reduce the number of network parameters by using depthwise separable convolution (DSC) and balance the detection accuracy and speed [38]. Subsequently, MobileNetV2 has added two features: the Inverted Residuals method makes the feature transmission more powerful, and the network layer is deeper [39]; the Linear Bottleneck module replaces the non-linear module and reduces the loss of low-level features.

MobileNetV3 released in 2019 combines part of the structures of V1 and V2 [40], integrates, optimizes, and deletes the network layers with high computational cost in the V2 system, and introduces the SE-Net (squeeze-and-excitation networks) lightweight attention structure without sacrificing accuracy while consuming low resources [41]. DSC (Depthwise Separable Convolution) consists of depthwise convolution (DW) and pointwise convolution (PW), as shown in Figure 6. Compared with traditional convolution, DSC reduces parameters and calculation greatly, and the comparison of calculations is shown in Formula (1):

$$\frac{w_1}{w_2} = \frac{D_k^2 \cdot M \cdot D_F^2 + M \cdot N \cdot D_F^2}{D_k^2 \cdot M \cdot N \cdot D_F^2} = \frac{1}{N} + \frac{1}{D_k^2} \tag{1}$$

where $w_1$ is the computational cost of depthwise separable convolution; $w_2$ is the computational cost of traditional convolution.

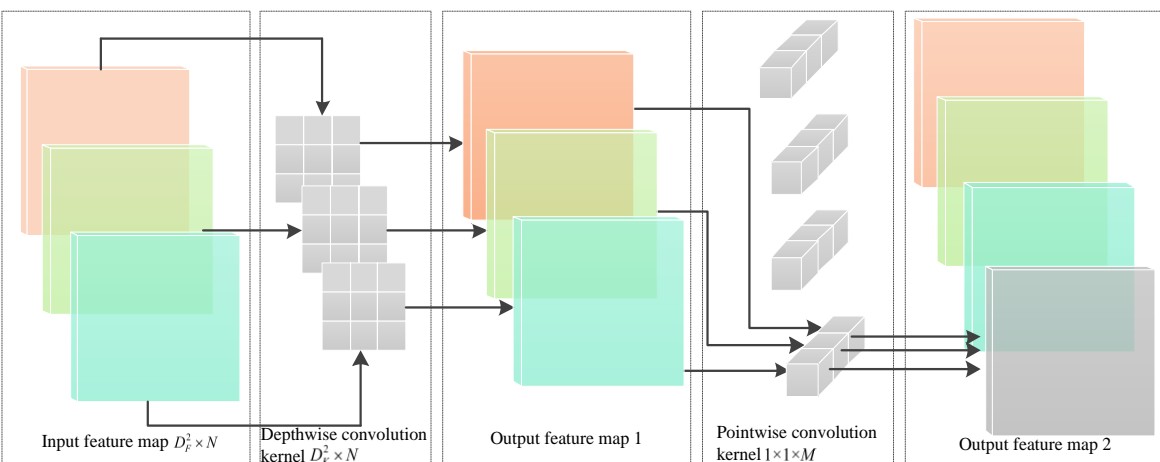

**Figure 6.** Principle of depthwise separable convolutions.

The MobileNetV3 feature extraction network adopts a 3 × 3 standard convolution and multiple inverted residual structures, which not only can reduce network parameters but also extract rich feature information. Among them, Figure 7a is the ResNet residual and Figure 7b is the reverse residual.

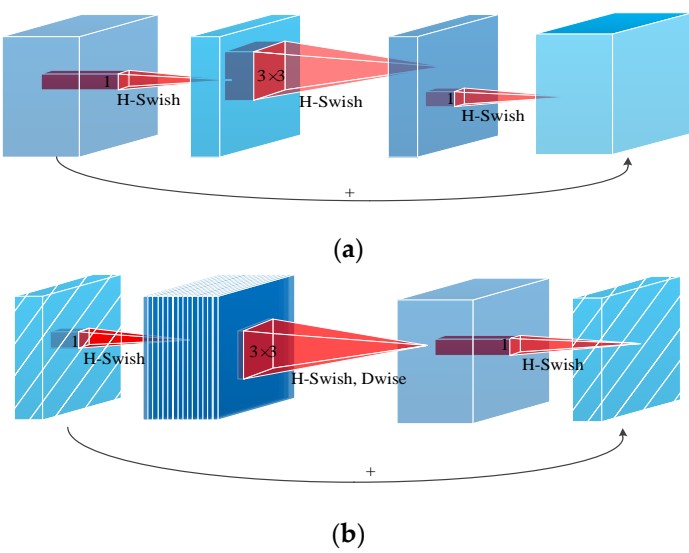

(**a**)

(**b**)

**Figure 7.** The residual block and inverted residual block. (**a**) The residual block. (**b**) The inverted residual block.

The reverse residual structure uses point convolution to increase the number of channels, then performs deep convolution at a higher level, and finally uses point convolution to reduce the number of channels. The reverse residual network improves the feature's gradient propagation ability with the help of residual connections, making the network layers deeper while using smaller input and output dimensions, greatly reducing the computational cost and parameter volume of the network. In addition, the reverse residual network has efficient CPU and memory inference capabilities, which can build flexible mobile models and thus be applicable to mobile device programs.

In MobileNet, two hyperparameters, $\alpha$ and $\beta$, are proposed. $\alpha$ is used as a width factor to adjust the number of convolution kernels to $\alpha$ times of the original one, and $\beta$ is used to control the size of the input image. The calculation of using DSC to adjust $\alpha$ is obtained by Formula (2):

$$W = D_K^2 \cdot \alpha M \cdot \beta D_F^2 + \alpha M \cdot \alpha N \cdot \beta D_F^2 \tag{2}$$

The adjustment of the width coefficient can directly reduce the computation and volume to $1/\alpha 2$, greatly reducing the number of model parameters and computation, with little loss of accuracy. In this paper, $\alpha$ is set to 0.5.

2.4.3. Convolutional Block Attention Module's Attention Mechanism

Most of the missed detections in the experiment occurred when the target size suddenly changed drastically, especially in the process of the bumpy field vehicle swaying left and right, the miss rate is very high. This also indirectly indicates that the native lightweight attention mechanism SE-Net in YOLOv5s may be limited in the case of sudden and drastic changes in target scale.

Compared with SE-Net, which only focuses on the importance of channel pixels, CBAM is a lightweight attention model that comprehensively considers the differences in importance between different channel pixels and the same channel pixels at different locations. It is a simple and efficient attention mechanism design, with minimum computational consumption, and it can be seamlessly integrated with convolutional networks and used for end-to-end training, as shown in Figure 8.

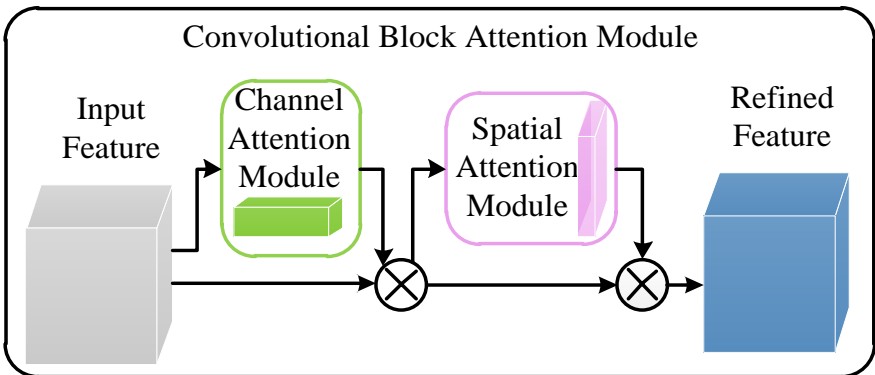

**Figure 8.** CBAM attention mechanism structure.

The CBAM consists of a channel attention module and a spatial attention module. The input features are inferred to contain attention features in sequence, and then the attention feature vector and the input feature vector are multiplied to achieve adaptive feature optimization. As shown in Figure 9, the channel attention vector is calculated along the spatial dimension to obtain the feature vector and is multiplied with the input feature. The channel attention mechanism is expressed in Formula (3):

$$M_c(F) = Sigmoid(MLP(AvgPool(F)) + MLP(MaxPool(F))$$
$$= Sigmoid(W_1(W_0(F_{avg}^C)) + (W_1(W_0(F_{max}^C)) \tag{3}$$

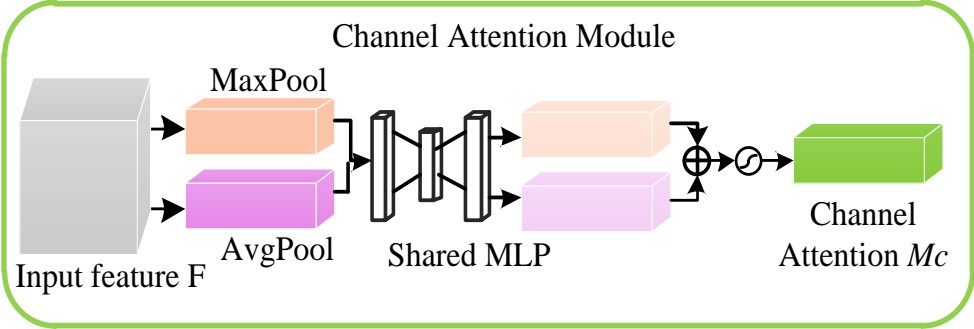

**Figure 9.** Channel attention mechanism.

Figure 10 represents the spatial attention vector, which is obtained by operating along the channel direction to obtain the feature vector and then multiplied by the input feature. The expression of the spatial attention mechanism is shown in Formula (4):

$$M_s(F) = Sigmoid(conv([AvgPool(F); MaxPool(F)]))$$
$$= Sigmoid(conv([F_{avg}^C, F_{max}^C])) \tag{4}$$
$$F' = M_S(M_C(F) \otimes F) \otimes (M_C(F) \otimes F)$$

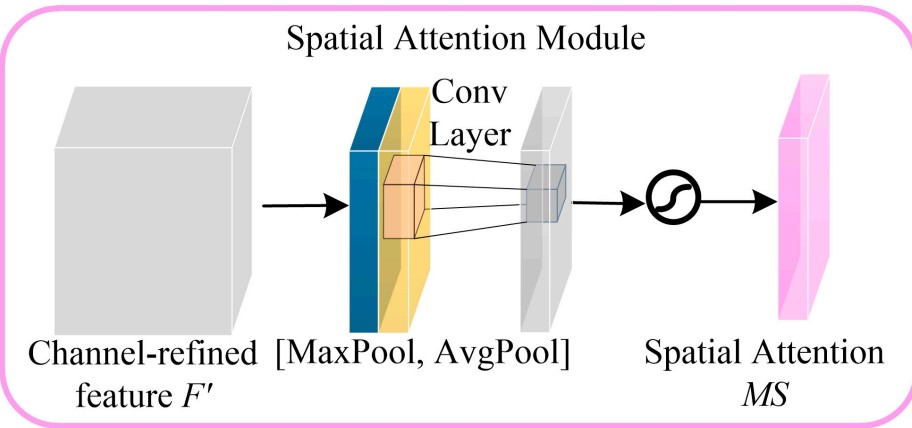

**Figure 10.** Spatial attention mechanism.

In the target detection model, replacing the SE-Net module with the CBAM attention mechanism to optimize the target detection accuracy makes the target feature extraction more complete, thus improving the target loss problem when the field bumping causes oscillating changes.

2.4.4. Improved Non-Maximum Suppression

When using Non-Maximum Suppression (NMS) to remove redundant detection boxes, the criterion for judging is the Intersection over Union (*IoU*) between the detection box and the box with the highest predicted score. When *IoU* is greater than the set threshold, the predicted detection box will be removed. However, in the case of densely distributed targets, due to the large overlap area of the detection boxes caused by the occlusion of the targets, the targets are often incorrectly removed by NMS, resulting in missed detection. Combining *DIoU* and NMS to improve missed detection, Distance-*IoU* Non-Maximum Suppression (*DIoU*-NMS) not only considers the value of *IoU*, but also considers the distance between the two box centers of the predicted boundary box and the true boundary box, as shown in Formula (5) of *DIoU*-NMS:

$$S_i = \begin{cases} S_i, & IOU - R_{DIOU}(M, B_i) < \varepsilon \\ 0, & IOU - R_{DIOU}(M, B_i) \geq \varepsilon \end{cases} \tag{5}$$

$M$ represents the prediction box with the highest predicted score; $B_i$ determines if the prediction box needs to be removed; $S_i$ is the classification score; $R_{DIOU}$ is the threshold for NMS; $R_{DIOU}$ represents the distance between the centers of the two boxes, as shown in Formula (6):

$$R_{DIOU} = \frac{\rho^2(b, b^{gt})}{c^2} \tag{6}$$

where $\rho^2(\cdot)$ represents the Euclidean distance; $b$ and $b^{gt}$ represent the distance between the predicted bounding box and the ground truth bounding box centers; $c$ represents the shortest diagonal length of the minimum bounding box encasing two boxes.

The biggest difference between *DIoU*-NMS and NMS is that when two boxes with far apart centers are encountered, *DIoU*-NMS considers the possibility that they may belong to different objects and should not be suppressed, thereby improving detection rates.

### 2.4.5. Knowledge Distillation

The technique of knowledge distillation is a widely adopted approach for compressing models, which differs from pruning and quantization in model compression. Essentially, knowledge distillation involves training a compact network model to emulate the knowledge extracted from a pre-trained larger network. This training methodology is commonly referred to as "teacher-student", where the larger network is the "teacher network", while the smaller network is the "student network". The goal of knowledge distillation is to enable the student network to achieve comparable, if not better, accuracy than the larger network while having fewer parameters and a smaller scale. By distilling the model, the problem of slow speed and high memory consumption is resolved, while also enhancing model accuracy. The distillation process can be observed in Figure 11.

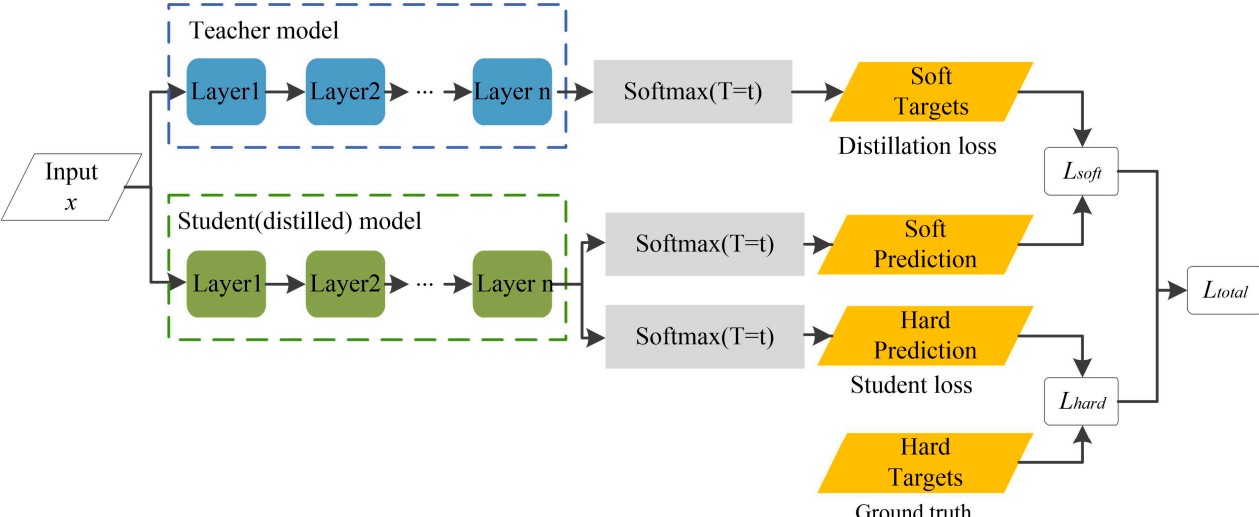

**Figure 11.** Knowledge distillation process.

This article utilizes the YOLOv5m model as the teacher model. Firstly, a deeper and more capable teacher network is trained using the data to extract features. Then, the teacher network outputs the logits function, which is distilled at a temperature of *T*. The class probability distribution obtained by applying the softmax layer is used as soft targets. At the same time, the student network outputs logits that are distilled at the same temperature *T*, and knowledge distillation is performed. This is a commonly used method for model compression, which differs from pruning and quantization. The main idea of knowledge distillation is to train a small network model to imitate a pre-trained large network. After the layer has been distilled, the class prediction probability is obtained as soft predictions, and the loss function is further obtained, shown in Formula (7):

$$L_{soft} = -\sum_{j}^{N} p_j^T \log q_j^T \tag{7}$$

where $p_j^T$ is the probability of predicting the *j*-th class using softmax at temperature *T* for the teacher network denoted; $q_j^T$ is the predicted probability of the *j*-th class using softmax at temperature *T* for the student network denoted.

Considering the teacher network also has a certain error rate, the loss function $L_{hard}$ is computed using the true labels as hard targets, combined with the original softmax output of the student network. The formula for $L_{hard}$ is shown in Formula (8):

$$L_{hard} = -\sum_j^N c_j \log q_j^T \tag{8}$$

where $c_j$ is the $j$th class ground truth label value.

$$L_{total} = \alpha L_{soft} + (1 - \alpha) L_{hard} \tag{9}$$

The final loss function $L_{total}$ is obtained by combining the weighted sum of the loss function $L_{hard}$ and $L_{soft}$, where $\alpha$ is the weighting coefficient.

The YOLOv5m model serves as the basis for the teacher model, which is initially trained to extract distinguishing features from the data. Usually, the teacher network has better classification or detection ability than the student network, and the more accurate the teacher network's classifications or detections are, the more beneficial it is for the student network's learning. Consequently, the student model acquires knowledge from the teacher model while cross-checking with the true labels, ensuring that it does not learn incorrect information.

### 2.5. Crop Row Fitting Method

2.5.1. Extraction of Crop Image ROI Based on Perspective Projection

In the actual navigation process of the weed control machine, the rows of crops are generally guided lines in the middle of the image, and the crops on the left and right sides closest to the guide line of the weed control machine directly affect the accuracy of the navigation. Therefore, pre-screening the ROI area in the image can reduce the amount of image data processing and reduce the interference of crop rows at the edge of the image, making the subsequent image processing steps more efficient. Therefore, it is necessary to construct the ROI of seedling images before training the YOLOv5 network model and then train the YOLOv5 network after labeling.

In order to determine the ROI of the crop rows in the image, the image is simplified to an imaging model under perspective projection, as shown in Figure 12. OO' is the center line of the crop rows in the middle of the image, EF and GH are the center lines of the left and right seedling rows, and the quadrangle ACDB is the ROI area.

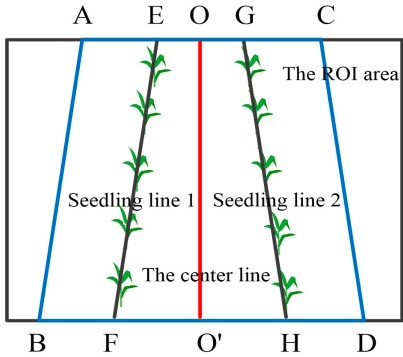

**Figure 12.** Simplified to an imaging model under perspective projection.

The specific process for determining ROI is as follows:

Under perspective projection, the pixel distance between the two ends of the central line of the maize crop on the DOG and DO'H is as follows (10):

$$G_G = \frac{D_{OG}}{D_{O'H}} = \frac{sin\left(\theta - \frac{\alpha}{2}\right)}{sin\left(\theta + \frac{\alpha}{2}\right)} \tag{10}$$

$G_G$ is the ratio of $D_{OG}$ to $D_{O'H}$; $\theta$ is the angle between the camera optical axis and a horizontal line, 45° to 60°; $\alpha$ is the vertical field of the view angle of the camera (°).

The pixel distance between point O' and point H can be obtained from the imaging of the camera.

$$D_{O'H} = \frac{2fd\sin\theta}{H(1+G_G)k} \tag{11}$$

$d$ is the average distance between rows of crops, mm; $f$ is the focal length of the camera, mm; $H$ is the height of the camera's optical center from the ground, mm; $k$ is the physical size of a single pixel in the image, mm.

When the camera is tilted to its maximum angle ($\theta = 60°$), the pixel distance $D_{O'H}$ between point O' and point H is the largest. By substituting A = 60° into Formula (9) and (10), the values of $D_{O'H}$ and $D_{OG}$ can be calculated. Using the referenced values of $D_{O'H}$ and $D_{OG}$, the area inside the quadrilateral ABCD is set as the ROI, and the non-ROI section of the image is masked and filled. Here, $D_{O'D} = D_{O'B} = cD_{MQ}$, $D_{OC} = D_{OA} = cD_{OG}$, and $c$ is the margin factor ($c$ = 1.2). The ROI extraction effect is shown in Figure 13.

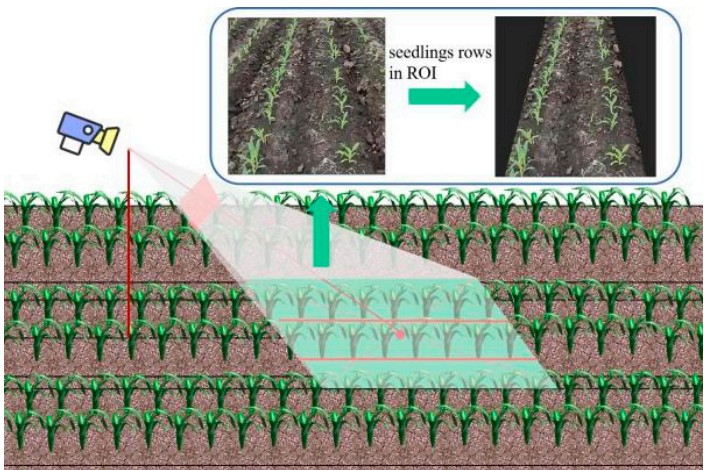

**Figure 13.** ROI projection extraction's result diagram.

2.5.2. Calculation of Maize Seedling Positions

As the maize seedling detection only provides bounding boxes with the coordinates of the box vertices, the maize positions are simplified to a coordinate value to represent their *x-y* positions in the image [42]. The bounding box position is described using coordinate information, starting from the vertex closest to the origin and moving clockwise. The definition of bounding box vertex positions is as follows: $(x_{\min}, y_{\min})$, $(x_{\max}, y_{\min})$, $(x_{\max}, y_{\max})$, and $(x_{\min}, y_{\max})$. In Formula (12), $x_{\min}$, $x_{\max}$, $y_{\min}$, $y_{\max}$ are used to define the positions of the maize seedlings, where $(x, y)$ represents the coordinates of the center of the guiding line for maize seedlings.

$$\begin{cases} x = (x_{\min} + x_{\max})/2 \\ y = (x_{\min} + x_{\max})/2 \end{cases} \tag{12}$$

2.5.3. Fitting of Crop Seedlings

The intelligent weed remover will adopt an automatic weeding system, taking into account that the crop rows in large fields usually do not deviate within a short distance. Moreover, since the tractor moves slowly at about 0.6 m/s, a simple linear fitting method can be used. Formulas (13)–(15) represent the seedling line fitting algorithm using the least squares method, and the center navigation line is fitted with linear fitting.

$$\begin{cases} \overline{X} = \frac{1}{n}\sum_{i=1}^{n} x_i \\ \overline{Y} = \frac{1}{n}\sum_{i=1}^{n} y_i \end{cases} \tag{13}$$

$$m = \frac{\sum_{i=1}^{n}(x_i - \overline{X})(y_i - \overline{Y})}{\sum_{i=1}^{n}(x_i - \overline{X})^2} \tag{14}$$

$$b = \overline{Y} - m\overline{X} \tag{15}$$

where $\overline{X}$ is calculated as the average of all x-coordinates of the points; $\overline{Y}$ is calculated as the average of all $y$-coordinates of the points.

The coordinates $(x_i, y_i)$ represent the location of each seedling on the graph, where $m$ represents the slope of the line and $b$ represents the y-intercept.

Crop line fit lines are indicated as in Formula (16):

$$y = mx + b \tag{16}$$

In order to separate the two crop rows, a threshold value $y_{th}$ and a reference point $s_c(x_c, y_c)$ are set, and the other points are defined as $x_i$, where $i \leq n$ is the sequence number for identifying crops and $n$ is the total number of points outside the reference point. Since the maize crop rows extend along the positive direction of the $y$-axis, these points need to be classified into different crop rows using only the $x_{ci} = x_c - x_i$ value for calculation. When $|x_{ci}| \leq x_{th}$, they are classified as L1 crop rows, and when $|x_{ci}| > x_{th}$, they are classified as L2 crop rows.

After classifying all detection points, Formulas (13)–(16) are used to calculate the m and b for L1 and L2 separately, and finally, the expressions for the two crop rows are obtained. The formulae are shown in (17):

$$\begin{cases} y_{L1} = m_1 x + b_1 \\ y_{L2} = m_2 x + b_2 \end{cases} \tag{17}$$

### 2.5.4. Calculation of Optimal Weed Removal Positions

In agricultural reclamation areas, the standardization of crop rows is high, and inter-row weeds are cleared by controlling the lateral displacement of the weeding shovel of the inter-row weeder. Building upon the previous Section 2.5.3 on crop row fitting, the next step involves determining the centerline of two crop rows. The diagram illustrating the position relationship between the weeding machine and the crop rows is shown in Figure 14. This can be achieved by obtaining the average of the fitted lines of the two crop rows to derive the navigation centerline, as shown in Formula (18):

$$\begin{cases} y = m_{center}x + b_{center} \\ m_{center} = \frac{m_1 + m_2}{2} \\ b_{center} = \frac{b_1 + b_2}{2} \end{cases} \tag{18}$$

where $m_1$ and $m_2$ are the slopes of the two crop rows, and $b_1$ and $b_2$ are the intercepts of the two crop rows.

We can express $h_x$ as a function of the position of the weeding machine and the fitted lines of the crop rows. Assuming the weeding machine position is $(x_{vehicle}, y_{vehicle})$, then $h_x$ is as shown in Formula (19):

$$h_x = \frac{|m_{center} \cdot x_{vehicle} - y_{vehicle} + b_{center}|}{\sqrt{1 + m_{center}^2}} \tag{19}$$

where $m_{center}$ is the slope of the central navigation line, and $b_{center}$ is the intercept of the central navigation line.

We can consider h as a function of the position of the weeding machine $(x_{vehicle}, y_{vehicle})$. Our goal is to find the position of the weeding machine that minimizes $h_x$.

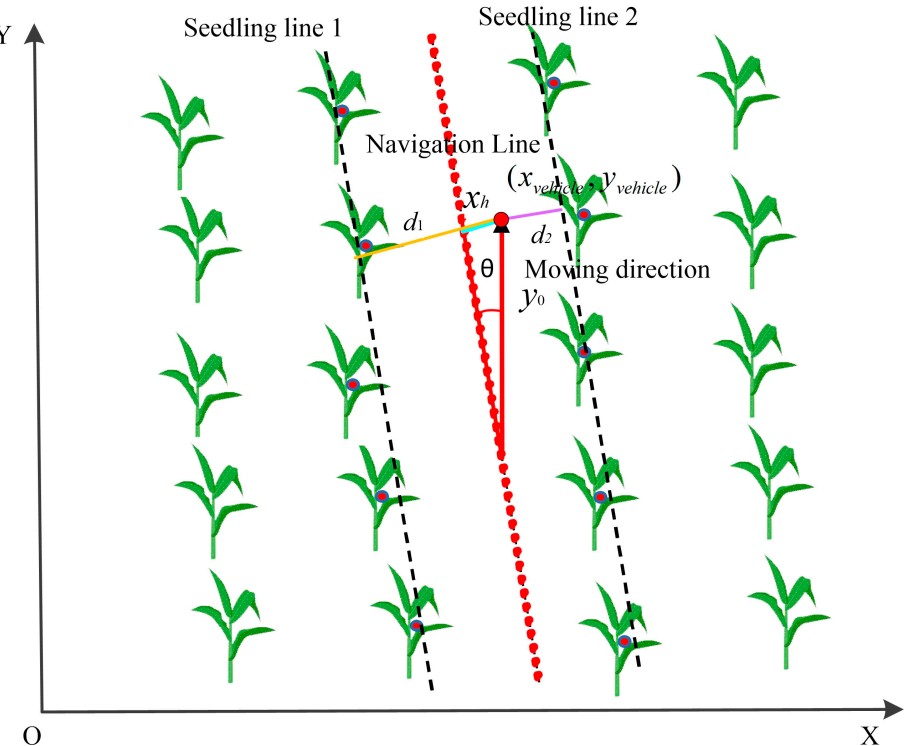

**Figure 14.** The position relationship between the weeding machine and the crop rows.

The navigation centerline is determined by the two crop rows to ensure that the weeding position follows the trajectory. Moreover, the greater the lateral distance between the two crop rows and the weeding shovel of the weeding machine, the lower the likelihood of damage to the maize leaves and roots. If we aim to maximize the proximity of the weeding machine's travel path to the two crop rows to ensure that the weeding machine is positioned as centrally as possible between the two rows of crops, we should use the minimum value to represent the distance from the weeding machine's travel path to the closer side of the two crop rows. This ensures that the vehicle does not deviate too far from the centerline and remains as close as possible to the middle of the two crop rows, maximizing the distance between the weeding machine's travel path and the two crop rows. For the two crop rows ($i = 1, 2$), we can calculate the perpendicular distance ($d_i$) from the weeding machine position to the crop rows, as shown in Formula (20). Our goal is to find the vehicle position that maximizes $\min(d_1 d_2)$.

$$d_i = \frac{|m_i \cdot x_{vehicle} - y_{vehicle} + b_i|}{\sqrt{1 + m_i^2}} \tag{20}$$

where $m_i$ is the slope of the $i$-th crop row, and $b_i$ is the intercept of the $i$-th crop row.

We can define the angle between the weeding machine's travel path and the fitted line of the crop row as $\theta$ and set the angle threshold as $\theta_{threshold}$. When the angle $\theta$ between the vehicle's travel path and the fitted line of the crop row reaches the set threshold $\theta_{threshold}$, we consider it as a curvature diagnosis, indicating the need for adjustment. The $\theta$ formula is shown in (21):

$$\theta = arctan\frac{x_h}{y_0} \tag{21}$$

where $y_0$ denotes the distance traveled by the weeding machine within the duration of the information update.

By comprehensively optimizing and considering three objectives, we can construct a comprehensive optimization function $F(x_{vehicle}, y_{vehicle})$, as shown in Formula (22):

$$F(x_{vehicle}, y_{vehicle}) = \alpha h_x(x_{vehicle}, y_{vehicle}) - \beta \min(d_1, d_2) - \gamma \max(0, |\theta - \theta_{threshold}|) \quad (22)$$

where $h_x(x_{vehicle}, y_{vehicle})$ represents the perpendicular distance from the weeding machine's travel path to the crop row; $\min(d_1 d_2)$ represents the maximum distance from the weeding machine's travel path to the two crop rows; $\alpha, \beta, \gamma$ are parameters that balance the three objectives.

This formula takes into account the minimization of $h_x$ and the maximization of the distance from the vehicle's travel path to the two crop rows, while considering the angle threshold and the constraints on perpendicular distance. Through numerical methods, we seek to find the vehicle position that maximizes $F(x_{vehicle}, y_{vehicle})$.

### 2.6. Data Collection and Preprocessing

In order to ensure the diversity of the dataset, the following factors were considered in the collection process. Different fields: the experiment involves three fields and the data were collected in 2022 from the fourth management area of Zhaoguang Farm in Beian City, Heilongjiang Province. When using the images of the dataset for maize seedling detection, the following factors were considered in the collection process in order to ensure the diversity of the dataset. Different plot conditions: mainly for the maize seedling period of the field scenes under the two sowing modes of no stubble stubble and stubble stubble, different interference factors, such as precipitation/accumulation, missing/row, weeds, canopy overlap, etc., affecting the detection are considered in the dataset. Weather factors are also taken into consideration, as well as natural light changes and shadows. Different physiological stages: the detection of maize seedlings is affected by the seedling growth cycle. This experiment selects different stages of continuous cycles from three-leaf to five-leaf stages. Unlike traditional seedling detection methods, the dataset does not contain images of individual maize seedlings. In addition, each image contains at least two rows of maize seedlings to improve robustness. The original dataset comprises 1500 images, with distribution as follows under different lighting conditions: 500 images captured under sunlight, primarily between 10 a.m. and 2 p.m.; 300 images taken under overcast conditions; 500 images acquired under strong light, mainly between 12 p.m. and 1 p.m.; and 200 images collected under weak light, primarily between 9 a.m. and 10 a.m., as well as between 3 p.m. and 4 p.m. Corresponding to different growth stages of the maize, the image count is as follows: 500 images for the three-leaf stage, primarily captured between 15 May and 25 May; 500 images for the four-leaf stage, mainly taken between 26 May and 5 June; and 500 images for the five-leaf stage, primarily acquired between 6 June and 15 June.

When manually collecting images, due to inconsistent shooting angles and the original image being too large, it would be too time-consuming to process the deep learning model, which cannot meet the required real-time performance. And due to the small sample dataset, the accuracy of target detection based on deep convolutional neural network is significantly related to the scale of sample dataset. In order to ensure the natural feature expression of the image and further improve the accuracy of the model, enlarging the number of training images can not only meet the requirements of deep networks and reduce the phenomenon of overfitting. Therefore, for the small-scale maize seedling image of this experiment, affine transformation, rotation clipping, flipping, and adding Gaussian noise are used to expand the dataset to 7000 images, further improve the model's perception of the image target position, and effectively extract features to optimize the performance of the network model. Then, the images are scaled to 416 pixels × 416 pixels according to the principle of proportional invariance. The original collected images and the enhanced samples are shown in Figure 15, and the dataset is randomly divided into training set, validation set, and test set according to the ratio of 8:1:1.

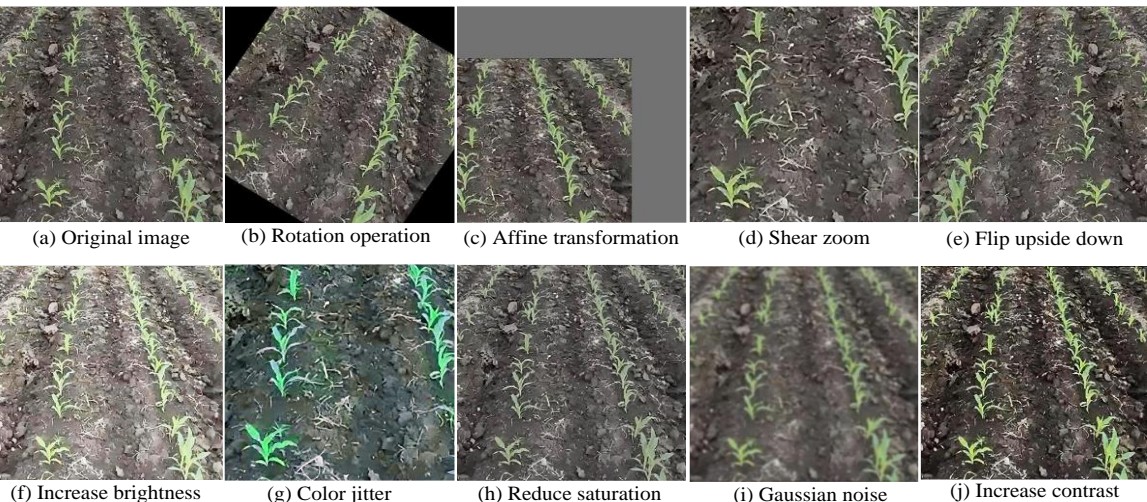

(a) Original image  (b) Rotation operation  (c) Affine transformation  (d) Shear zoom  (e) Flip upside down

(f) Increase brightness  (g) Color jitter  (h) Reduce saturation  (i) Gaussian noise  (j) Increase contrast

**Figure 15.** Sample data of maize seedlings.

### 2.7. Experimentation and Analysis

#### 2.7.1. Experimental Platform and Parameter Settings

In the process of model training, stochastic gradient descent (SGD) is used as the optimizer, the momentum factor is set to 0.937, the initial learning rate is 0.01, and the learning rate is adjusted by cosine annealing. The weight decay coefficient is set to 0.0005, the batch size is set to 32, and the training epoch is set to 200. All experiments in this experiment were conducted on the experimental platform of Table 2. When training a deep learning model for small-scale data samples, introducing transfer learning can reduce overfitting and speed up model convergence, improving training effect. Therefore, all the models established in this paper are loaded with pre-trained weights based on the voc 2012 dataset in the training process.

**Table 2.** Experimental platform.

| Name | Device-Related Configuration |
| --- | --- |
| CPU | 11th Gen Intel(R) Core(TM)i7-11700@2.50 GHz |
| Main memory | 16 GB |
| GPU | NVIDIA GeForce GTX 1080 Ti |
| GPU acceleration library | CUDA11.0.3, CUDNN8.2.1 |
| Operating system | Windows 10 (64 bit) |
| Software environment | Python 3.7, Pytorch 1.7.0 |

#### 2.7.2. Model Evaluation Metrics

The experimental evaluation of the model's overall performance involves the selection of the following metrics: mean average precision (*mAP*) for object detection accuracy, frames per second (*FPS*) for detection speed, precision-recall curve (*P-R* curve) for precision and recall assessment, harmonic mean *F1*-score, floating point operations (*FLOPs*) for computational complexity, and total training parameters (*Params*) for model size evaluation.

The precision (*P*) represents the proportion of true positive samples among all the samples predicted as positive. The calculation formula is given by Formula (23):

$$P = \frac{TP}{TP + FP} \times 100\% \tag{23}$$

where *TP* is the number of true positive samples; *FP* is the number of false positive samples; *FN* is the number of false negative samples; *N* is the number of predicted classes.

The recall (*R*) represents the proportion of samples predicted as positive among all the true positive samples. The calculation formula is given by Formula (24):

$$R = \frac{TP}{TP + FN} \times 100\% \tag{24}$$

The *AP* value is the area between the precision-recall curve and the coordinate axes. The calculation formula is given by Formula (25):

$$AP = \int_0^1 P(R)d_R \tag{25}$$

Mean average precision (*mAP*) can comprehensively evaluate the localization and classification effects of the model for multiple categories and multiple targets. Calculating *mAP* requires calculating the *AP* (average precision) of each category in the recognition task, and then taking its average, the formula of which is (26):

$$mAP = \sum \frac{AP_i}{N} \tag{26}$$

where *N* is the total number of classes; $AP_i$ refers to the *AP* value of the *i*-th class.

As precision (*P*) and recall (*R*) are complementary to each other, this paper evaluates the experimental results by using the harmonic mean *F1* value of the two, as given by calculation Formula (27):

$$F1 = \frac{2 \times P \times R}{PR} \tag{27}$$

*FLOPs* are the floating-point operations used to compute the network model, which evaluate the time complexity of the model. The calculation formula is given by Formula (28):

$$FLOPs = 2 \times (C_{out} \times H_{out} \times W_{out} \times C_{in} + C_{out}) \tag{28}$$

$H_{out}$ and $W_{out}$ refer to the height and width of the output feature map; $C_{in}$ and $C_{out}$ refer to the input and output channel numbers.

*Params* is the total number of parameters that need to be trained in the network model, which corresponds to the consumption of hardware memory resources and is used to evaluate the space complexity of the model. The calculation formula is given by Formula (29):

$$Params = k^2 \times C_{in} \times C_{out} + C_{out} \tag{29}$$

where *k* is the convolution kernel size.

### 3. Results

*3.1. Test Results of Various Backbone Networks*

Based on the empirical data delineated in Table 3, a comprehensive analysis of MolieNetv3 as the backbone network within the YOLOv5s framework has been conducted, with a comparative scrutiny against other prevalent networks.

**Table 3.** Results of YOLOv5s different backbone networks.

| Network Model | Backbone Network | F1-Score/% | Params/$10^6$ | FLOPs/$10^9$ | mAP/% | FPS/(frame·s$^{-1}$) |
|---|---|---|---|---|---|---|
| YOLOv5s | CSPDarkNet-53 | 90.2 | 7.21 | 7.5 | 89.4 | 31 |
| | EfficientNet | 88.3 | 3.62 | 7.1 | 86.2 | 35 |
| | DensenNet-169 | 87.4 | 14.2 | 33.1 | 86.9 | 17 |
| | ResNet-50 | 88.9 | 25.6 | 10.3 | 87.3 | 27 |
| | ShuffleNetV2 | 86.1 | 3.12 | 5.9 | 85.2 | 30 |
| | MolieNetv3 | 91.2 | 5.42 | 6.2 | 91.8 | 33 |

Performance metrics reveal that MolieNetv3 achieved an *F1*-score of 91.2% and an *mAP* of 91.8%. These figures surpass those of peer architectures such as CSPDarkNet-53, EfficientNet, DensenNet-169, ResNet-50, and ShuffleNetV2, unequivocally demonstrating MolieNetv3's exceptional performance in object detection accuracy and reliability.

In terms of model scale and computational requisites, MolieNetv3's parameterization is a mere $5.42 \times 10^6$, significantly lower than that of CSPDarkNet-53 at $7.21 \times 10^6$, DensenNet-169 at $14.2 \times 10^6$, and ResNet-50 at $25.6 \times 10^6$. This notable reduction in model parameters affords MolieNetv3 a distinct advantage in terms of model lightweighting. Additionally, MolieNetv3's computational efficiency is highlighted by its *FLOPs* count of $6.2 \times 10^9$, which is substantially more favorable when compared to the *FLOPs* of CSPDarkNet-53, DensenNet-169, and ResNet-50.

On the front of resource consumption, the high efficiency of MolieNetv3 implies its capability to deliver high-performance object detection in resource-constrained environments, such as edge computing devices or mobile platforms, without exerting excessive pressure on the device's battery life or computational resources.

### 3.2. Ablation Experiment

Based on the experimental results in Table 4, when MolieNetv3 is used as the backbone network embedded in the YOLOv5s framework, the model maintains a parameter count of 5.42 million, *FLOPs* of $6.2 \times 10^9$, *mAP* of 91.8%, and a model file size of 11.2 MB. Upon introducing CBAM, the model's parameter count and *FLOPs* remain unchanged, but the *mAP* increases to 92.2%, with the model file size staying at 11.3 MB. With the addition of *DIoU*-NMS, the parameter count and *FLOPs* remain unchanged, yet the *mAP* further improves to 92.3%, while the model file size remains the same. Upon introducing $L_{\text{soft}}$, the parameter count and *FLOPs* decrease further to 3.21 million and $5.1 \times 10^9$, respectively, while the *mAP* remains at 92.2%, and the model file size decreases to 7.5 MB.

**Table 4.** Ablation experimental results.

| Model | MolieNetv3 | CBAM | *DIoU*-NMS | $L_{\text{soft}}$ | *Params*/M | *FLOPs*/$10^9$ | *mAP* | Model File/MB |
|---|---|---|---|---|---|---|---|---|
| | - | - | - | - | 7.21 | 7.5 | 89.4 | 14.5 |
| | √ | - | - | - | 5.42 | 6.2 | 91.8 | 11.2 |
| YOLOv5s | √ | √ | - | - | 5.42 | 6.2 | 92.2 | 11.3 |
| | √ | √ | √ | - | 5.42 | 6.2 | 92.3 | 11.3 |
| | √ | √ | √ | √ | 3.21 | 5.1 | 92.2 | 7.5 |

In summary, based on the experimental results in Table 4, it can be concluded that in the YOLOv5s framework, utilizing MolieNetv3 as the backbone network and introducing components such as CBAM, *DIoU*-NMS, and $L_{\text{soft}}$ can significantly enhance the performance metrics (e.g., *mAP*) of the object detection model, while effectively reducing the model's parameter count, *FLOPs*, and model file size. These results indicate that the proposed YOLOv5s model based on MolieNetv3 demonstrates high performance and efficiency in object detection tasks, making it suitable for deployment and application in resource-constrained environments.

Based on the confusion matrix plots of YOLOv5s and YOLOv5-M3 in Figure 16, which are used for classifying images into three categories: maize, weed, and background, the following observations can be made:

For YOLOv5s (a): the diagonal elements indicate high correct prediction values for each category: maize (0.89), weed (0.90), and background (0.72), suggesting good classification performance. The off-diagonal elements, particularly background being predicted as mazie (0.1) and weed being predicted as background (0.28), indicate misclassification errors. The misclassification rate between mazie and weed is relatively low, but the misclassification rate for background is high. For YOLOv5-M3 (b): the diagonal elements also show high values, indicating good performance: mazie (0.91), weed (0.95), and background (0.75),

which is better than that of YOLOv5s. The off-diagonal elements show misclassifications: background being predicted as mazie (0.08) and weed being predicted as background (0.25), similar to the issues in the left matrix, but with a slight decrease in the misprediction of maize. Comparing the two, YOLOv5-M3 seems to perform slightly better overall, with higher true positive rates (diagonal elements) and slightly lower misclassification rates. The most significant difference is in the predictions involving the background category, where YOLOv5s has a higher false positive rate for predicting background as maize compared to YOLOv5-M3.

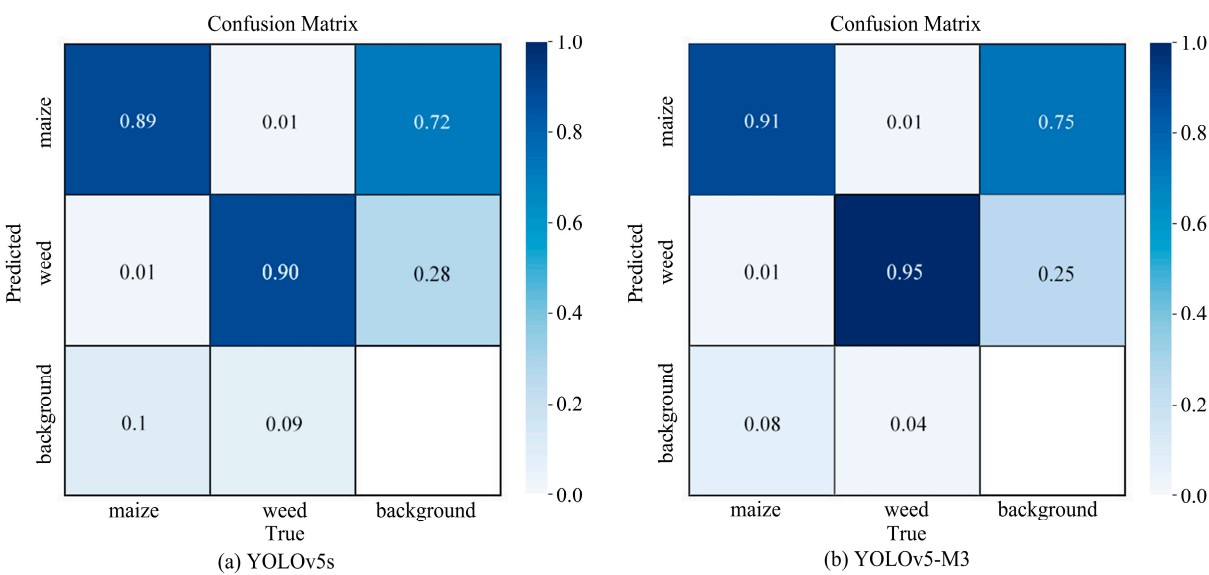

**Figure 16.** Confusion matrix plots.

### 3.3. Test Results of Different Network Models

After a comparative analysis of the test results of different network models in Table 5, we can draw the following conclusions: in terms of precision, YOLOv5-M3 leads with a result of 93.2%, indicating its superior capability in accurately identifying targets. Following closely is YOLOv5s with a precision of 91.2%, while Faster-RCNN and SSD are at 85.1% and 86.9%, respectively. As for the recall rate, Faster-RCNN ranks first with 87.8%, suggesting it performs best in minimizing the omission of true targets. YOLOv5-M3 follows with a recall rate of 91.1%, with YOLOv5s and YOLOX slightly behind at 89.2% and 88.1%, respectively.

**Table 5.** Effect comparison of different network models.

| Network Model | Precision/% | Recall/% | F1-Score/% | Params/$10^6$ | FLOPs/$10^9$ | mAP | Model File/MB | FPS/(frame·s$^{-1}$) |
|---|---|---|---|---|---|---|---|---|
| Faster-RCNN | 85.1 | 87.8 | 86.4 | 136 | 18.5 | 86.9 | 89.3 | 0.45 |
| YOLOv5s | 91.2 | 89.2 | 90.2 | 7.21 | 7.5 | 89.4 | 14.5 | 23 |
| SSD | 86.9 | 85.7 | 86.3 | 33.2 | 8.9 | 86.3 | 92.1 | 11 |
| YOLOX | 89.2 | 88.1 | 88.6 | 8.93 | 4.5 | 88.7 | 17.1 | 50 |
| YOLOv5-M3 | 93.2 | 91.1 | 92.1 | 3.21 | 5.1 | 92.2 | 7.5 | 39 |

The trade-off between precision and recall is reasonable, as a higher recall rate means more weeds might be misidentified as seedlings, but it reduces the rate of missed seedlings. In contrast, higher precision ensures fewer weeds are detected as seedlings. Considering the large-scale field planting of maize, where the distance between two adjacent seedlings does not have significant positional deviation, the accuracy of maize seedling detection has a minimal impact on the extraction of guiding lines. However, a higher recall rate could introduce errors in the guiding lines. Therefore, adjusting to reduce the recall rate

minimizes the excessive deviation of the weeding machinery, avoiding damage to the root system of maize seedlings.

The *F*1-score, a harmonic mean of precision and recall, takes into account both the accuracy and robustness of the model. On this metric, YOLOv5-M3 also ranks first with a score of 92.1%, demonstrating its superior overall performance. YOLOv5s has an *F*1-score of 90.2%, higher than SSD and Faster-RCNN, which are at 86.3% and 86.4%, respectively.

In terms of the number of model parameters and computational complexity (*FLOPs*), YOLOv5-M3 shows significant advantages with only 321 million parameters and $2.13 \times 10^9$ *FLOPs*, which is far less than the 13.6 billion parameters and $18.5 \times 10^9$ *FLOPs* of Faster-RCNN. This indicates that YOLOv5-M3 maintains high precision while being more lightweight and computationally efficient.

The *mAP* is an important indicator of the overall performance of a detection model. YOLOv5-M3 leads with a score of 92.2%, indicating the highest average level of detection accuracy across different thresholds. Regarding model file size, YOLOv5-M3 also shows a significant advantage, with a size of only 8.5MB, which is very beneficial for deployment on resource-constrained devices. Finally, in terms of frames per second (*FPS*), YOLOX leads with a processing speed of 50 frames per second, indicating its strong capability in real-time processing. YOLOv5-M3 also has a respectable processing speed of 39 frames per second.

In summary, YOLOv5-M3 exhibits outstanding performance in precision, recall rate, *F*1-score, model lightweightness, and processing speed. It is especially advantageous in model efficiency and practicality, making it highly suitable for deployment in real-world applications.

*3.4. Improved Testing of YOLOv5s Network Model*

The dataset also includes images of irregularly shaped maize seedlings or weeds. In the collected images, the distribution of crops and weeds is shown in Figure 17, mainly divided into four situations: weed-free areas, dense weed distribution, sparse weed distribution, and maize seedling missing areas. To demonstrate the superiority of our proposed model in detecting weeds and confirm the effectiveness of our improved YOLOv5-M3 network model, four typical images were chosen from various environments where weeds are dense, sparse, distant from crops, or multiple weed species coexist. A comparison was made with four classic models, namely Faster-RCNN, SSD, and YOLOX, and the detection results are presented in Figure 17b–e.

For YOLOv5s (a): the diagonal elements indicate high correct prediction values for each category: maize (0.89), weed (0.90), and background (0.72), suggesting good classification performance. The off-diagonal elements, particularly background being predicted as mazie (0.1) and weed being predicted as background (0.28), indicate misclassification errors. The misclassification rate between mazie and weed is relatively low, but the misclassification rate for background is high. For YOLOv5-M3 (b): the diagonal elements also show high values, indicating good performance: mazie (0.91), weed (0.95), and background (0.75), which is better than that of YOLOv5s. The off-diagonal elements show misclassifications: background being predicted as mazie (0.08) and weed being predicted as background (0.25), similar to the issues in the left matrix, but with a slight decrease in the misprediction of maize. Comparing the two, YOLOv5-M3 seems to perform slightly better overall, with higher true positive rates (diagonal elements) and slightly lower misclassification rates. The most significant difference is in the predictions involving the background category, where YOLOv5s has a higher false positive rate for predicting background as maize compared to YOLOv5-M3.

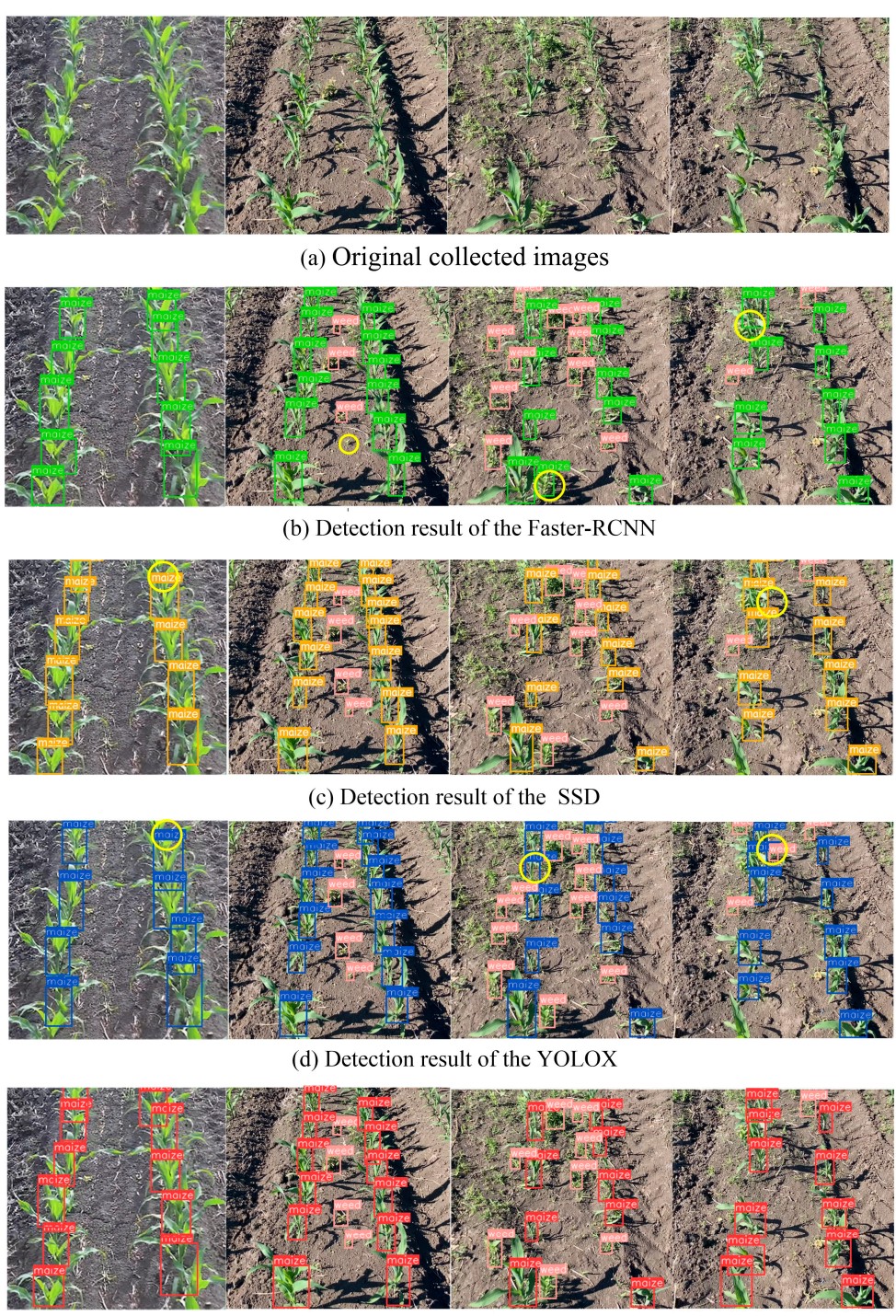

(a) Original collected images

(b) Detection result of the Faster-RCNN

(c) Detection result of the SSD

(d) Detection result of the YOLOX

(e) Detection result of the YOLO-M3

**Figure 17.** Detection results of models. Note: Yellow circles represent false detections and missed detections.

## 3.5. Crop Row Fitting Accuracy

By identifying and extracting regions of interest (ROI) from field images, precise location information of maize seedlings can be obtained, allowing for the extraction of crop center point coordinates. Utilizing these coordinates, the least squares method is applied to generate two fitting lines, with the median line serving as the desired navigation line, which is essential for guiding agricultural machinery with precision. To validate the adaptability of the least squares method in various weed environments, we selected four typical weed distribution scenarios for testing the fitting accuracy. By comparing the angle

error between manually annotated center points and the navigation line obtained from least squares fitting, we can assess the fitting accuracy. Root mean square error (RMSE) is a commonly used metric for measuring such errors, and a fitting is considered suboptimal when the RMSE exceeds 5°.

According to the data presented in Table 6, it is observed that in densely weeded environments, the average angle error is 3.13°, with a processing time of only 65 milliseconds. This indicates that the least squares method can provide relatively accurate results at high speeds, even in complex environments. When the weed distribution is sparse, the fitting accuracy improves, with the average angle error reduced to 2.03° and the processing time slightly decreasing to 62 milliseconds. Moreover, when weeds are distant from crops or when multiple weed species coexist, the fitting accuracy is further enhanced, achieving an average angle error of 1.32°, and the processing time is reduced to 53 milliseconds.

**Table 6.** Fitting accuracy results of crop row navigation lines.

| Four Situations | Average Angular Deviation (°) | Execution Time (ms) |
|---|---|---|
| Maize seedling missing | 3.13 | 51 |
| Dense weed distribution | 3.91 | 65 |
| Sparse weed distribution | 2.43 | 62 |
| Weed-free | 2.32 | 53 |

These data suggest that the least squares method not only performs excellently in complex environments with dense weed distribution but also maintains efficiency and accuracy in other weed conditions. Therefore, the least squares method is highly suitable for agricultural machinery navigation tasks, as it can provide the precision required for practical applications while ensuring speed. This offers robust technical support for automated machinery navigation in precision agriculture, contributing to improved operational efficiency and refined crop management levels.

## 4. Discussion

The primary limitations of deep learning in plant detection include reliance on substantial amounts of labeled data, poor performance in small sample learning, and weak model generalization capabilities [43]. The depthwise separable convolutions can effectively reduce model complexity while maintaining accuracy and increasing detection speed, making it suitable for plant disease classification tasks [44]. The improved YOLOv5-M3 model provides crucial support for the classification and localization of these targets, which is vital for precision agriculture and crop management. The model achieved an average detection accuracy of 91.4% and increased the frame rate to 39 frames per second, demonstrating its practical feasibility for real-time applications. The method proposed in this paper, compared to the moving window scanning method used by [18], utilizes an end-to-end deep learning framework, offering higher identification efficiency without depending on the regularity of crop rows. Compared to the Hough transform-based method proposed by [45], our method does not require a preset starting point for crop rows and exhibits greater robustness, capable of identifying both regular and irregular curved and straight crop rows. In contrast to the morphological feature-based method proposed by [46], our approach does not depend on the physiological structure of crops and is not limited to single plants, allowing it to process images of entire rows of crops. Compared to the Xception model used by [47], the YOLOv5-M3 model employed in this paper has fewer parameters, faster recognition speed, and is more suitable for real-time applications.

The incorporation of an attention mechanism in the feature extraction backbone network of YOLOv4 facilitated the detection of weeds in carrot fields. However, YOLOv3/v4 is computationally intensive, resulting in slow recognition speeds and posing challenges for deployment on embedded devices. In contrast, the YOLOv5 utilized in this study has lower computational requirements and faster recognition speeds [48]. The CNN

method relies on a large amount of annotated data, and its performance in small sample learning is poor. The original YOLOv5 method performs poorly in recognizing small samples and complex backgrounds [34]. This paper improves the learning ability of small samples through data augmentation and incorporates modules such as CBAM to enhance recognition capability, resulting in better recognition performance. In an effort to balance precision and computational efficiency for embedded applications, this research introduces the utilization of CBAM, which enhances feature extraction capabilities, particularly beneficial for identifying small objects and complex backgrounds. The model employs *DIoU*-NMS in lieu of the traditional NMS [49]. This approach considers the distance between bounding box centers, which not only retains targets more effectively but also distinguishes overlapping boxes with greater precision. This results in improved accuracy and recall rates while minimizing the interference between objects and reducing false positives. During the knowledge distillation process, the student model learns from both the teacher model and the true labels, effectively preventing the propagation of incorrect information from the teacher model and enhancing the overall quality of the model [50]. In summary, the model design meticulously considers the balance between lightweight efficiency and recognition accuracy, theoretically making it more suitable for the dataset scenario at hand, thereby achieving commendable detection results.

Building on the current research, future studies could focus on further enhancing the real-time detection accuracy of the YOLOv5-M3 model, potentially by integrating additional lightweight attention mechanisms or loss functions to boost performance. It is essential to further investigate the adaptability and resilience of the algorithm under varying environmental conditions and weed densities. Additionally, future research could delve deeper into the practical implications of error analysis on navigation lines, particularly in terms of fulfilling agricultural tasks and the operational requirements of autonomous machinery.

## 5. Conclusions

The present study innovatively applies deep learning-based object detection methods to the automatic weeding system for maize, aiming to improve identification accuracy and ensure effective weed control while addressing the issue of mechanical weeding causing damage to maize seedlings and roots.

(1) The improved YOLOv5s network model utilizes MobileNetV3 for feature extraction to enhance the lightweight approach of the original YOLOv5s and introduces the lightweight attention mechanism CBAM and focal loss function to improve the feature extraction capability of the detection model. During the knowledge distillation process, in addition to learning from complex teacher network models, the lightweight model is also compared with real labels, effectively preventing erroneous information from being distilled into the lightweight network from the teacher network, thus endowing the lightweight model with stronger learning ability. The YOLOv5-M3 model significantly reduces the missed detection rate, while also optimizing model size and detection speed to enhance its generalization and robustness.

(2) The improved YOLOv5s network model provides technical support for the classification and localization of maize seedling targets. Experimental results demonstrate that the proposed model achieves an average detection accuracy of 92.2%, which is comparable to the YOLOv5s network model, with an increased frames per second of 39.

(3) Comparative experiments with Faster-RCNN, SSD, YOLOv4, and YOLOX in terms of detection accuracy, computational complexity, parameter quantity, and detection speed validate the superiority and effectiveness of the YOLOv5-M3 model. The model's strong adaptability and anti-interference capability are verified under varying weed conditions. The error analysis of the manually labeled centerline and navigation midline demonstrates an error of less than 5°, meeting the practical operational requirements.

**Author Contributions:** W.Z. and H.G. conceived the study and designed the project. H.G. performed the experiment, analyzed the data, and drafted the manuscript. X.W. helped to revise the manuscript. All authors have read and agreed to the published version of the manuscript.

**Funding:** This research was funded by the National Key Research and Development Program, grant number 2016YFD020060802, the Heilongjiang Provincial Key R&D Program, grant number 2023ZXJ07B02, and the "Three Verticals" Basic Cultivation Program, grant number ZRCPY202306.

**Institutional Review Board Statement:** Not applicable.

**Data Availability Statement:** Data are contained within the article.

**Conflicts of Interest:** The authors declare no conflicts of interest.

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
