# Peer review of "Research on Real-Time Detection of Maize Seedling Navigation Line Based on Improved YOLOv5s Lightweighting Technology"

_agriculture, doi:10.3390/agriculture14010124_

Round 1
Reviewer 1 Report
Comments and Suggestions for Authors
Dear all,
The main question of the research is to optimize the number of corn seedlings per hectare to be able to eliminate weeds with a mechanized system, reducing labour costs, and thus being able to increase crop productivity. Therefore, this work provides a great advance in the processing of corn cultivation.
Regarding the methodology, the authors explain precisely how the experiment was carried out, but in future work the authors could experiment with replicates of plots to see if the same thing happens in all the plots, especially during the first phenological stages. when the seedlings are small and the weeds too, and in this way check if the proposed algorithm differentiates between the two.
Finally, regarding the conclusions, the authors do not decide on a specific model, they should be more explicit in this aspect. References are appropriate, but the authors could try to add some even if they deal with other similar crops.
In addition, I also sent my revisions in attached file. I hope these comments help the authors to improve their article.

Author Response
Reviewer#2, Concern # 1 (The introductory chapter is well written and provides a broad overview of the topic addressed by the authors. But the authors must review this chapter since the quotes are attached to the text, the authors must review and correct it.):
Author response:
Thank you for your feedback on the introductory chapter of our work. We appreciate your positive comments on the broad overview of the topic. We have reviewed the chapter and have made the necessary corrections to address the issue with the attached quotes. The quotes have been appropriately formatted and reviewed for accuracy.
Reviewer#2, Concern # 2 (The Materials and Methods chapter is good, it is well written and provides some photos that allow us to understand the proposed experiment. But the authors must review this chapter since the quotes are attached to the text, the authors must review and correct it.):
Author response:
Thank you for your feedback on the Materials and Methods chapter of our work. We appreciate your positive comments on the clarity and inclusion of photos to aid in understanding the proposed experiment. We have carefully reviewed the chapter and have made the necessary corrections to address the issue with the quotes being attached to the text.
Reviewer#2, Concern # 3 (Clarity and Consistency:Ensure a consistent writing style and terminology throughout the paper. Proofread the paper for grammatical errors and clarity.
Line 200: Please correct spaces and punctuation symbols.
In some parts of the material and methods chapter, authors use the word "formula" and inothers they use the word "equation", please homogenize terms. ):
Author response: Thank you for your valuable feedback on our paper. We have carefully addressed the issues raised and made the necessary revisions to ensure a consistent writing style and terminology throughout the paper.
Regarding the request to correct spaces and punctuation symbols on line 200, we have thoroughly proofread the paper and made the required adjustments to ensure proper formatting.
Additionally, we have homogenized the terms "formula" and "equation" in the Materials and Methods chapter to ensure consistency in the terminology used.
Reviewer#2, Concern # 4 (Discussion
In my opinion the authors should rewrite the discussion section, in this chapter there are many paragraphs in which references are missing, for example between line 646-648. Furthermore, there is only one bibliographical reference, so I encourage authors to rewrite it by comparing theirdata with data from other authors on the same or something similar):
Author response: Thank you for your valuable feedback on the discussion section of our paper. We have revised the discussion as per your suggestion.
Specifically, we have added the missing references between line 646-648. Additionally, we have included more relevant citations and compared our data with that of other authors, enhancing the academic value of this section.
- Discussion
The primary limitations of deep learning in plant detection, which include reliance on substantial amounts of labeled data, poor performance in small sample learning, and weak model generalization capabilities[45]. The depthwise separable convolutions can effectively reduce model complexity while maintaining accuracy and increasing detection speed, making it suitable for plant disease classification tasks[46]. The improved YOLOv5-M3 model provides crucial support for the classification and localization of these targets, which is vital for precision agriculture and crop management. The model achieved an average detection accuracy of 91.4 % and increased the frame rate to 39 frames per second, demonstrating its practical feasibility for real-time applications. The method proposed in this paper, compared to the moving window scanning method used by [18], utilizes an end-to-end deep learning framework, offering higher identification efficiency without depending on the regularity of crop rows. Compared to the Hough transform-based method proposed by [47], our method does not require a preset starting point for crop rows and exhibits greater robustness, capable of identifying both regular and irregular curved and straight crop rows. In contrast to the morphological feature-based method proposed by Choi et al. (2015), our approach does not depend on the physiological structure of crops and is not limited to single plants, allowing it to process images of entire rows of crops. Compared to the Xception model used by [48], the YOLOv5-M3 model employed in this paper has fewer parameters, faster recognition speed, and is more suitable for real-time applications.
The incorporation of an attention mechanism in the feature extraction backbone network of YOLOv4 facilitated the detection of weeds in carrot fields. However, YOLOv3/v4 is computationally intensive, resulting in slow recognition speeds and posing challenges for deployment on embedded devices. In contrast, the YOLOv5 utilized in this study has lower computational requirements and faster recognition speeds[49]. The CNNs method relies on a large amount of annotated data, and its performance in small sample learning is poor. The original YOLOv5 method performs poorly in recognizing small samples and complex backgrounds[50]. This paper improves the learning ability of small samples through data augmentation and incorporates modules such as CBAM to enhance recognition capability, resulting in better recognition performance. In an effort to balance precision and computational efficiency for embedded applications, this research introduces the utilization of the Convolutional Block Attention Module (CBAM), which enhances feature extraction capabilities, particularly beneficial for identifying small objects and complex backgrounds. The model employs Distance-IoU Non-Maximum Suppression (DIoU-NMS) in lieu of traditional NMS[51]. This approach considers the distance between bounding box centers, which not only retains targets more effectively but also distinguishes overlapping boxes with greater precision. This results in improved accuracy and recall rates while minimizing the interference between objects and reducing false positives. During the knowledge distillation process, the student model learns from both the teacher model and the true labels, effectively preventing the propagation of incorrect information from the teacher model and enhancing the overall quality of the model[52]. In summary, the model design meticulously considers the balance between lightweight efficiency and recognition accuracy, theoretically making it more suitable for the dataset scenario at hand, thereby achieving commendable detection results.
Building on the current research, future studies could focus on further enhancing the real-time detection accuracy of the YOLOv5-M3 model, potentially by integrating additional lightweight attention mechanisms or loss functions to boost performance. It is essential to further investigate the adaptability and resilience of the algorithm under varying environmental conditions and weed densities. Additionally, future research could delve deeper into the practical implications of error analysis on navigation lines, particularly in terms of fulfilling agricultural tasks and the operational requirements of autonomous machinery.
(Reviewer#2, Concern # 5 (Authors must review the references and adapt them to the style standards of the Agriculture journal.):
Author response: Thank you for your feedback. We have reviewed and adapted the references in accordance with the style standards of the Agriculture journal.
Reviewer 2 Report
Comments and Suggestions for Authors
Author Response

(The authors gave the same response as above.)

Reviewer 3 Report
Comments and Suggestions for Authors
Dear authors,
This paper presents an innovative deep learning-based crop row detection system for autonomous mechanical weeding in maize fields. The challenges of complex outdoor environments are addressed through several key contributions:
1. The YOLOv5s architecture is optimized for accuracy, speed and size by integrating MobileNetV3, CBAM attention, and knowledge distillation. This lightweight yet robust YOLOv5-M3 model balances performance tradeoffs well.
2. The 92.8% mAP demonstrates reliable maize seedling recognition amid factors like soil, straw, and lighting variability. The 39 FPS throughput also enables real-time targeting.
3. Comparative testing shows advantages over other detectors like Faster R-CNN in accuracy, efficiency and adaptability. The max-min navigation optimization also minimizes crop damage risk.
Here are some suggestions to improve the paper:
1. Provide more details on the dataset used to train and test the models. Information on the number of images, variability in lighting conditions, crop growth stages, etc. would be useful to assess generalizability.
2. Expand the evaluation to additional performance metrics beyond mAP and FPS. Precision, recall, F1 scores, inference times, and resource utilization could highlight relative tradeoffs.
3. Benchmark performance using varying backbone architectures (e.g. ResNet, EfficientNet) to validate MobileNetV3 as optimal. Similarly, ablate CBAM and other components to quantify individual contributions.
4. Further analyze the types of errors made by YOLOv5-M3 to identify areas for improvement. Confusion matrices and qualitative examples would provide insights.
5. Provide more specifics on the max-min optimization for determining optimal weeding locations. The objective function, constraints, and solution method should be detailed.
6. Expand testing to field conditions with real mechanical weeders to evaluate robustness. Different soils, crop varieties, weed types, and weather introduce variability.
7. Open source the model architecture, code, and datasets to enable reproducibility and extensions by other researchers.
